# Non-Equilibrium Living Polymers

**DOI:** 10.3390/e22101130

**Published:** 2020-10-06

**Authors:** Davide Michieletto

**Affiliations:** 1School of Physics and Astronomy, University of Edinburgh, Peter Guthrie Tait Road, Edinburgh EH9 3FD, UK; davide.michieletto@ed.ac.uk; 2MRC Human Genetics Unit, Institute of Genetics and Molecular Medicine, University of Edinburgh, Edinburgh EH4 2XU, UK

**Keywords:** living polymers, DNA, topology, non-equilibrium polymer physics

## Abstract

Systems of “living” polymers are ubiquitous in industry and are traditionally realised using surfactants. Here I first review the theoretical state-of-the-art of living polymers and then discuss non-equilibrium extensions that may be realised with advanced synthetic chemistry or DNA functionalised by proteins. These systems are not only interesting in order to realise novel “living” soft matter but can also shed insight into how genomes are (topologically) regulated in vivo.

## 1. Introduction

Topological constraints among polymers determine the viscoelastic properties of complex fluids and soft materials such as creams, oils, gels and plastics that we use everyday. During the last decades several theories have been proposed to connect the macroscopic and rheological behaviour of complex fluids with the microscopic properties of entangled polymers. Among the most successful, there are the reptation and tube theories and their extensions, such as constraint release or double-reptation [1]. One of the strongest assumption of these theories is that polymer chains do not change their architecture, length or topology within experimental or practical timescales. One important class of systems that do not obey this constraint is the family of so-called “living” polymers [2,3,4].

Living polymers are different from their non-living counterparts as random architectural changes such as breakage, fusion and reconnections alter the architecture and topology of the polymers on timescales shorter than, or comparable to, their relaxation. Models of living polymers have been successfully applied to explain the behaviour of certain surfactants that form worm-like micelles [5]. One crucial feature of living polymers is that they are fundamentally in equilibrium (unless external forces are applied) and that the architectural re-arrangements occur randomly at any point and time along the polymers’ contours. Importantly, changing the structure of the polymers on timescales shorter than their own (reptation) relaxation brings about intriguing rheological behaviours such as dramatic shear thinning, banding or even thickening [6,7].

In this paper, I first review the theoretical state-of-the-art of traditional living polymers and then discuss potential non-equilibrium generalisations. It should be stressed that these systems of “active” polymers are different from the ones studied recently in which the energy is transformed into translational motion [8,9] and instead mostly focus on architectural and topological alterations to the polymers’ structure. Finally, I will discuss potential realisations of these systems using DNA mixed with proteins and enzymes that are commonly found in vivo and that contribute to the (topological) regulation of genomes.

### 1.1. Review of Equilibrium Living Polymers

Living polymers with dynamic architecture can be realised using certain surfactants that create micelles: structures made of amphiphilic molecules that combine a water-loving, or hydrophilic, part with a water-hating, or hydrophobic, parts. When embedded in aqueous solutions amphiphilic molecules form hollow structures (the micelles) in which the hydrophobic part is shielded from the water. In certain regimes of surfactant and salt concentrations, micelles can self-assemble to form elongated structures with a self-explanatory name of “worm-like” micelles (Figure 1A). These structures take the form of effective polymers, which can become entangled with each other and hence confer viscoelasticity to the solution. Standard polymers relax mainly by reptation, i.e., a slithering motion of the polymer within the tube formed by neighbouring polymers (Figure 1B); instead, worm-like micelles can also break, fuse and reconnect with neighbours and hence form a “living” network of entanglements with unique rheological properties (Figure 1C).

The stress relaxation of a chain can be computed as the survival probability of the average tube segment. A segment of the tube that is present at time t=0 disappears when the chain travels far enough and one of its termini crosses one of the boundaries of the tube (Figure 1B). This problem can be recast into that of a diffusing tube along a static polymer and the monitored segment can be represented as a 1D Brownian walk on a domain of length *L* with absorbing boundary conditions [1,2]. For standard, monodisperse polymer systems relaxing by reptation, the stress relaxation function, i.e., the response of the system to an infinitesimal perturbation, is essentially the survival probability of a diffusing particle placed at random on a 1D interval of length *L* and with absorbing boundaries; it can be written as [1]
(1)$$\mu \left(t\right)=\frac{8}{\pi^{2}}\sum_{p=odd}p^{-2}e^{-p^{2}t/T_{d}}$$
where $T_{d}\left(L_{0}\right)=L_{0}^{2}/\left(D_{c}\pi^{2}\right)$ is the relaxation (or “reptation”) time, $D_{c}∼D_{0}/L_{0}$ is the diffusion coefficient of the centre of mass of the polymer (and that of the particle) and $D_{0}$ a microscopic diffusion constant.

A theory for the relaxation of “living polymers” must also account for the reversible breakage and fusion and it was first proposed by Cates and co-authors [2,3,4,10]. This theory assumes that the system is in equilibrium with respect to the breakage/fusion process and that breakage can occur at any point along the polymer contour. Because of this morphological process the system attains polydispersity in lengths with mean $L_{0}$. For exponentially polydisperse polymers, the stress relaxation function is proportional to the following survival function
(2)$$\mu \left(t\right)=\frac{1}{L_{0}}\int_{0}^{\infty }e^{-L/L_{0}}e^{-t/T_{d}\left(L\right)}dL$$
where $P\left(L\right)=e^{-L/L_{0}}/L_{0}$ is the distribution of polymer lengths with mean $L_{0}$ and $e^{-t/T_{d}\left(L\right)}$ is the longest contribution (p=0 component) of the reptative relaxation for a polymer of length *L*. The solution to Equation (Equation 2) can be found via a saddle-point approximation to be a stretched exponential with exponent 1/(β+1), with β=3 for reptation dynamics [2,11,12].

From the survival function μ(t), the stress relaxation can be found as $G\left(t\right)=G_{0}\mu \left(t\right)$, with $G_{0}$ an instantaneous shear modulus. In turn, the zero-shear viscosity (which will be used later on) can be computed as
(3)$$\eta_{0}=\int_{0}^{\infty }G\left(t\right)dt\;.$$

One of the key timescales in the system is the breakage timescale $\tau_{b}={\left(\kappa L_{0}\right)}^{-1}$ where κ is the number of breaks per unit time per unit length. Since the system is also in morphological equilibrium, the timescale for the fusion process must be the same as the breakage one. The kinetics of breakage and fusion are effectively implemented by introducing dynamical and discontinuous changes in the positions of the absorbing boundaries at a certain rate (see Figure 2A). This stochastic process can no longer be mapped to a diffusive equation and it cannot be exactly solved analytically [2]. The relevant adimensional quantity is now the ratio of the breakage and reptation timescales, i.e.,
(4)$$\xi \equiv \frac{\tau_{b}}{T_{d}}≃\frac{D_{0}}{\kappa L_{0}^{4}}\;.$$

If ξ≫1 the breakage dynamics is slower than the relaxation of the chains and hence reptation dominates, whereas if ξ≤1 the breakage dynamics occurs on timescales comparable, or shorter than, the reptative ones. In this case the reaction process—which can take place anywhere along the polymers—has the net effect of “democratising” the stress relaxation of the micelles and forces it to become closer to a simple exponential (rather than a stretched one) in spite of the sample polydispersity. The explanation for this near-Maxwellian, or simple exponential, stress relaxation of polydisperse micelles is one of the fundamental results of this theory (see Figure 2B).

In this regime, one may then ask what is the typical relaxation timescale, accounting for the architectural change. To compute this, one should notice that for a breakage to be “useful” in accelerating the relaxation of the chain it has to happen within a distance λ from the current position of the diffusing particle (representing the relaxing tube segment) such that the particle will on average cross the new boundary (and hence be absorbed) before a new fusion event occurs. This can be calculated simply as
(5)$$\lambda^{2}≃D_{c}\tau_{b}≃\frac{D_{0}}{L_{0}}\frac{1}{\kappa L_{0}}=L_{0}^{2}\xi \;.$$
The rate limiting step thus involves waiting for a breakage to happen within a distance λ from the relaxing segment, which happens on average at a rate
(6)$$\kappa_{r}≃\kappa \lambda ≃\kappa L_{0}\xi^{1/2}≃\tau_{b}^{-1}\xi^{1/2}$$
or at timescale
(7)$$\tau_{r}=\kappa_{r}^{-1}≃T_{d}\xi^{1/2}≃{\left(T_{d}\tau_{b}\right)}^{1/2}\;.$$

We can then find that the viscosity depends on the parameters of the system as
(8)$$\eta_{0}=\int_{0}^{\infty }G\left(t\right)dt=G_{0}\tau_{r}≃G_{0}{\left(D_{0}\kappa \right)}^{-1/2}L_{0}\;,$$
which depends only linearly on the average chain length, rather as $L_{0}^{3}$ valid for unbreakable chains (see Figure 2D).

A numerical algorithm to simulate equilibrium living polymers is detailed in Reference [2] and also implemented as a C++ code by the author and shared in a git repository (see acknowledgement section); briefly, one should simulate the diffusion of a particle deposited at random along a 1D interval with exponentially distributed length (and mean $L_{0}$). The diffusion depends on the instantaneous length as $D_{0}/L$ and breakages or fusions of an *l*-long segment can occur at rate κ per unit time and unit length and at rate $\kappa e^{-l/L_{0}}$ per unit time and per each end, respectively (see also Figure 2A). The survival function μ(t) is computed as the probability of a particle to have not reached one of the two ends of the interval by time *t*. Examples of this function for different values of ξ are given in Figure 2B. The adimensional viscosity is the suitably normalised integral of μ(t) and is plotted in Figure 2C as a function of ξ for fixed $L_{0}=1$—thus confirming Equation (Equation 7)—and in Figure 2D as a function of the mean length $L_{0}$ for unbreakable chains and living polymers thereby confirming Equation (Equation 8).

### 1.2. Non-Equilibrium Living Polymers

Starting from the standard theory for equilibrium living polymers introduced in Reference [2] and reviewed above, there are several out-of-equilibrium generalisations that can be proposed and studied. Here, I propose to study systems in which *only* breakage or fusion are considered; they can yield systems with architectural absorbing states in which all chains have been broken or fused together. As we shall see, I argue that these systems are interesting not only because they can be readily realised experimentally but also because they display intriguing rheological regimes.

#### 1.2.1. Breakage Only

In the case of pure breakage the key adimensional quantity is best expressed as $\chi =\xi^{-1}=T_{d}/\tau_{b}$, as it is the number of breakage events per reptation time. If χ>1 then multiple break points are introduced along the tube within one reptation time. In general, this system is out-of-equilibrium and so the average segment length depends on how much time it has passed since the start of the experiment, as
(9)$$\ell \left(t\right)=\frac{L_{0}}{n_{c}\left(t\right)+1}=\frac{L_{0}}{\left(\chi t/T_{d}+1\right)}\;,$$
where $n_{c}\left(t\right)=\chi t/T_{d}$ is the average number of “cuts” in time ($n_{c}\left(t\right)+1$ is the number of segments). In turn, the typical relaxation timescale at time *t* is the one necessary for the relaxation of the average segment length with instantaneous curvilinear diffusion $D_{c}=D_{0}/\ell \left(t\right)$, i.e.,
(10)$$\tau_{r,break}=\frac{\ell {\left(t\right)}^{2}}{D_{c}\left(t\right)}=\frac{\ell^{3}\left(t\right)}{D_{0}}≃\frac{L_{0}^{3}}{D_{0}{\left(\chi t/T_{d}+1\right)}^{3}}$$
which emphasises that (i) the relaxation of the system changes in time, (ii) for small ageing times (up to the typical breakage time $\tau_{b}=T_{d}/\chi$) one recovers the behaviour of unbreakable polymers $T_{d}∼L_{0}^{3}/D_{0}$ and (iii) at ageing times larger than the breakage time (but smaller than the one for which reptation is no longer a good model for the dynamics of the segments) one finds that
(11)$$\tau_{r,break}∼\frac{1}{D_{0}\kappa^{3}t^{3}}\;.$$

This ageing behaviour, i.e., dependence of relaxation time on the age of the sample, is peculiar of out-of-equilibrium systems. One way to monitor this ageing is to compute the age-dependent survival function
(12)$$\eta \left(t;T_{a}\right)=\int_{T_{a}}^{T_{o}+T_{a}}\mu \left(t;T_{a}\right)dt$$
where $T_{a}$ is the age of the sample and $T_{o}$ the observation time.

The numerical implementation of this “breakage-only” case is straightforward and can be done as follows (source codes are available at a git repository, see acknowledgement section): starting from the algorithm for equilibrium living polymers (as explained in Figure 2A) one needs to disallow “fusion” events and to start the simulation from exponentially polydisperse segments with mean $\ell \left(t\right)=L_{0}/\left(n_{c}\left(t\right)+1\right)$. In order to compute an “instantaneous” viscosity at age $T_{a}$ I choose to compute the survival function $\mu (t;T_{a})$ over an observation period $T_{o}=100T_{d}$. The results are plotted in Figure 3 where it is shown that indeed $∼T_{a}^{-3}$ captures the decay in viscosity for $T_{a}/T_{d}>1/\chi$ well and that is instead roughly constant at small ageing times.

It is worth mentioning that this set-up can be realised with DNA and certain type of enzymes; I discuss these potential experiments in more detail below.

#### 1.2.2. Fusion Only

On the contrary, for pure fusion, the relevant adimensional quantity is $\phi =T_{d}/\tau_{f}$, i.e., how many fusion events happen before the longest relaxation time of the chain ($\tau_{f}$ is the typical time in between fusion events). Once again, the complex fluid is expected to age; in particular, for ϕ≥1 the relaxation is expected to be much longer than the reptation time already at short times whereas for ϕ<1 the fluid is initially relaxing as a normal polymeric system of unbreakable chains and increasing the relaxation time at later ageing time. The instantaneous relaxation time can be estimated as the time required by any one segment to reach one of the termini of the tube, accounting for the fact that new tubes—on average, $L_{0}$ long—are added to the original tube at rate $\tau_{f}^{-1}$ (this is valid only for a fusing test chain in a reservoir of chains that cannot fuse between them, see below for a more self-consistent argument). One may thus argue that the typical length of the tube at time *t* is on average
(13)$$\ell \left(t\right)=\left(n_{f}\left(t\right)+1\right)L_{0}$$
where $n_{f}\left(t\right)=\phi t/T_{d}$ is the average number of fusion events in time (and $n_{f}+1$ is the number of segments of mean length $L_{0}$). The relaxation time is thus
(14)$$\tau_{r,fuse}\left(t\right)=\frac{\ell {\left(t\right)}^{2}}{D_{c}}≃\frac{L_{0}^{3}{\left(\phi t/T_{d}+1\right)}^{3}}{D_{0}}\;.$$

As before, at small times (shorter than the typical fusion time $\tau_{f}=T_{d}/\phi$) we recover the relaxation time of unbreakable chains $\tau_{r,fuse}∼T_{d}≃L_{0}^{3}/D_{0}$ whereas at longer times ($t>\tau_{f}$) the fluid displays ageing and the viscosity should increase as
(15)$$\tau_{r,fuse}∼T_{d}\tau_{f}^{-3}t^{3}$$
until all chains have been fused together.

The calculation above is correct if one considers one test chain that can fuse with others and that is embedded within a reservoir of chains that cannot fuse among them. For a system in which all chains can fuse together, it underestimates the real growth of a test chain as one would instead need to add segments that are polydisperse with growing mean ℓ(t). In this case the growth of a test chain is exponential, i.e.,
(16)$$\ell \left(t\right)=L_{0}e^{t/\tau_{f}}\;,$$
and, in turn, the relaxation time also becomes exponential
(17)$$\tau_{r,fuse}=\frac{\ell {\left(t\right)}^{2}}{D_{c}}∼\frac{L_{0}^{3}e^{3t/\tau_{f}}}{D_{0}}\;$$
which again shows that at short times ($t<\tau_{f}$) the relaxation is expected to follow the one for non-living chains, while at large times the viscosity is expected to diverge exponentially at we thus expect a gel or glassy behaviour, i.e., one for which the relaxation function μ(t) never decays to zero within experimental or numerical timescales.

Once again, the numerical implementation is straightforward (codes available at a git repository, see acknowledgement section) as it is possible to simulate each observation window of time (which I here take to be $T_{o}\gg T_{d}$) at a certain ageing time $T_{a}$ starting from a situation in which polymers are exponentially polydisperse with mean $\ell \left(T_{a}\right)=L_{0}\left(\phi T_{a}/T_{d}+1\right)$. During the observation time the chains can fuse at a rate $\tau_{f}^{-1}$ and relax via reptative mechanisms with instantaneous diffusion coefficient $D_{c}=D_{0}/\ell \left(t\right)$ but cannot break. Viscosity curves for different values of ϕ and as a function of the ageing time are reported in Figure 4.

It should be mentioned once more that since there is no counterpart to each architectural change (e.g., breakage-only systems do not have fusion events, and vice versa), the systems are driven to architectural absorbing states in which all polymers in the sample are either broken down to individual segments (for breakage-only) or fused in a giant sample-spanning structure (for fusion only). Once the absorbing state is reached, no more breakage or fusion events can occur and the systems are therefore stuck. In both these cases, the relaxation times are expected to be independent on the initial average length $L_{0}$. Additionally, in the case of pure breakage, it is expected that when the breaking of chains brings the mean length below about one entanglement length, the reptation theory does not apply any longer for the relaxation of the segments.

Finally, Equations (Equation 11) and (Equation 15) (or (Equation 17)) describe markedly different behaviours that can be tuned by suitable choice of the “architectural” timescales $\tau_{b}$ and $\tau_{f}$. These two behaviours are also decoupled in non-equilibrium systems and so can be independently tuned while simultaneously acting on the system. Given this enhanced control over the rheological properties of systems of (non-equilibrium) living polymers, one may thus wonder whether these peculiar systems may find practical use. To this end, in the next section I discuss possible experimental realisations.

### 1.3. Non-Equilibrium DNA Digestion and Ligation

Systems of non-equilibrium living polymers may be realised using DNA functionalised by certain classes of proteins. For instance, using restriction enzymes (RE) it is possible to cut—or “digest”—DNA thereby generating irreversible breakages along the contour. More specifically, (type II) REs recognise specific DNA sequences and break the sugar-phosphate DNA backbone in correspondence of those sequences [13]. This process is akin to the “breakage-only” process described above with the caveat that typically DNA molecules are (i) initially monodisperse in length and (ii) have a finite number of restriction sites along their contour. For this reason at large times one expects the viscosity to reach a plateau at about $\eta_{0}\left(t\to \infty \right)/\eta_{0}\left(t=0\right)∼N_{RS}^{-3}$, where $N_{RS}$ is the number of restriction sites and if the large time regime is still well described by reptation.

It should also be noted that the time for a typical digestion experiment is about 15 min in optimal conditions and high-fidelity enzymes or ∼ hours for less ideal conditions [13]. At this time all the restriction sites available have been cleaved. Thus, the typical cutting time of one restriction site is of the order of minutes or tens of minutes, considering that a typical RE has between 1 and 10 sites per DNA molecule (of course this number depends on the specific combination of DNA sequence and RE considered). This cutting time should be compared with the typical relaxation time of, for example, λ−DNA in moderately entangled conditions, which is of the order to 1–10 s (at 0.5–5 mg/mL) [14]. Thus, the regime that can be attained in experiments that can be performed with moderately entangled λ—DNA is that in which χ<1 (as the one numerically reported in Figure 3) so that at small ageing times the solution should behave as a standard complex fluid and display deviations from this behaviour only at large ageing times (following the scaling of Equation (Equation 10)).

It is also worth noting that type II REs leave so-called “sticky ends” in correspondence of the cleaved sequences. These sticky ends can in principle re-anneal by thermal fluctuations but can never permanently fuse back two segments. In order to achieve irreversible fusion of DNA molecules one needs ligase enzymes (and ATP) [15]. Using this family of proteins one can in principle also reproduce the condition of irreversible fusion described above and so induce a gelling by exponential growth of the polymers in solution.

Finally, it should be highlighted that enzymes such as restriction and ligase are commonly found in vivo where they fulfil important biological functions. For instance, restriction enzymes are found in bacteria and used as a defence mechanism against viral infection of phages. At the same time, ligases catalyse the formation of a phosphodiester bond and are required to repair DNA single or double-strand breaks in vivo. Other proteins that change the architecture and topology of DNA, such as Topoisomerase [16] or Structural Maintenance of Chromosome (SMC) complexes [17,18], may also be used to create new viscoelastic regimes of entangled DNA in vitro. Understanding these regimes will also shed light into how the topology of genomes are regulated in vivo, where DNA is stored under extreme conditions of confinement and crowding.

## 2. Conclusions

In this work I have reviewed the standard theory for equilibrium living polymers [2,3,4,10] and proposed simple non-equilibrium generalisations. I have focused in more detail to the case of irreversible breakage and fusion and derived scaling laws for the change of zero-shear viscosity as a function of ageing time.

I have also discussed how these systems may be realised experimentally using solutions of DNA functionalised by enzymes that are used in routine molecular biology experiments such as restriction enzymes and ligases. While in the literature some groups have studied the change in rheological behaviour of solutions of linear and ring DNA subject to the action of Topoisomerase [19,20,21], here I have theoretically considered other classes of proteins, e.g. restriction and ligation enzymes, that may be expected to yield interesting non-equilibrium rheological regimes by altering DNA’s architecture.

The challenge of achieving a comprehensive understanding of the (linear and non-linear) rheological behaviours of these non-equilibrium complex fluids will certainly stimulate both theoreticians and experimentalists in the near future.

## Figures and Tables

**Figure 1 entropy-22-01130-f001:**
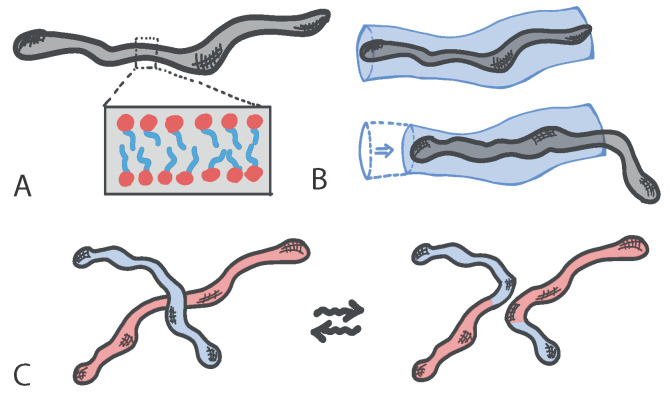
(**A**) Worm-like micelles. In inset, the disposition of amphiphilic molecules to shield the hydrophobic part from the aqueous solution. (**B**) Relaxation of a tube segment by reptation. (**C**) Relaxation of two worm-like micelles via a reversible architectural reconnection.

**Figure 2 entropy-22-01130-f002:**
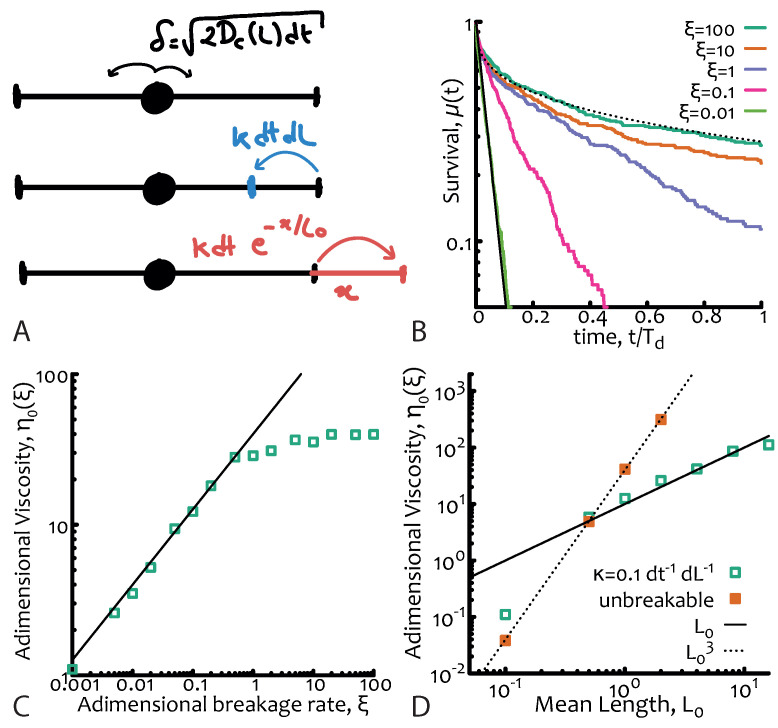
**Equilibrium living polymers.** (**A**) Sketch of the numerical algorithm. From top to bottom: (i) diffusion step (showing the explicit dependence on instantaneous length) (ii) breakage event and (iii) fusion event with respective rates. Absorbing conditions are set at the boundaries of the interval. (**B**) Survival function μ(t) for different values of the adimensional parameter $\xi =\tau_{b}/T_{d}$. Dotted and solid lines are stretched and normal exponential, respectively and are drawn as a guide for the eye. (**C**) Corresponding zero-shear viscosity showing the dependence $\eta_{0}∼\xi^{1/2}$ for ξ≲ 1 (black line). (**D**) Scaling of the adimensional viscosity with mean length $L_{0}$ showing that is, as expected, $∼L_{0}$ for living polymers and $∼L_{0}^{3}$ for unbreakable ones. Curves in (**B**) and data points in (**C**), (**D**) are obtained from 1D Brownian simulations using the numerical algorithm described in (**A**) with parameters $L_{0}$ = 1, dL=0.01, dt=0.01 and $D_{0}=0.01$ dL${}^{2}$/dt.

**Figure 3 entropy-22-01130-f003:**
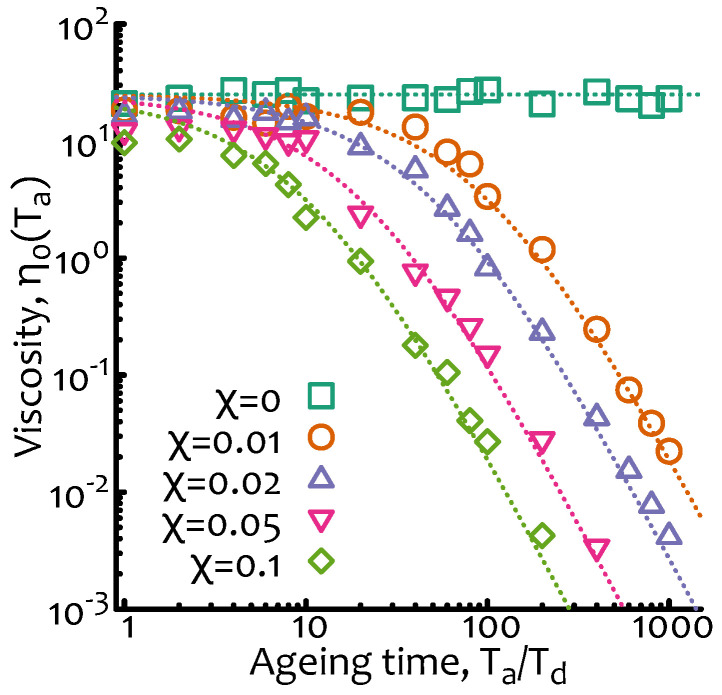
**Non-equilibrium living polymers—breakage only.** Zero-shear viscosity as a function of the ageing time $T_{a}/T_{d}$ and for different values of $\chi =T_{d}/\tau_{b}$. Dashed lines are obtained from Equation (Equation 10) multiplied by a numerical pre-factor. The data points are from Brownian simulations as described in Figure 2A with no fusion and with parameters $D_{0}=0.01$ dL${}^{2}$/dt, dL=0.01, dt=0.01 and $L_{0}=1$.

**Figure 4 entropy-22-01130-f004:**
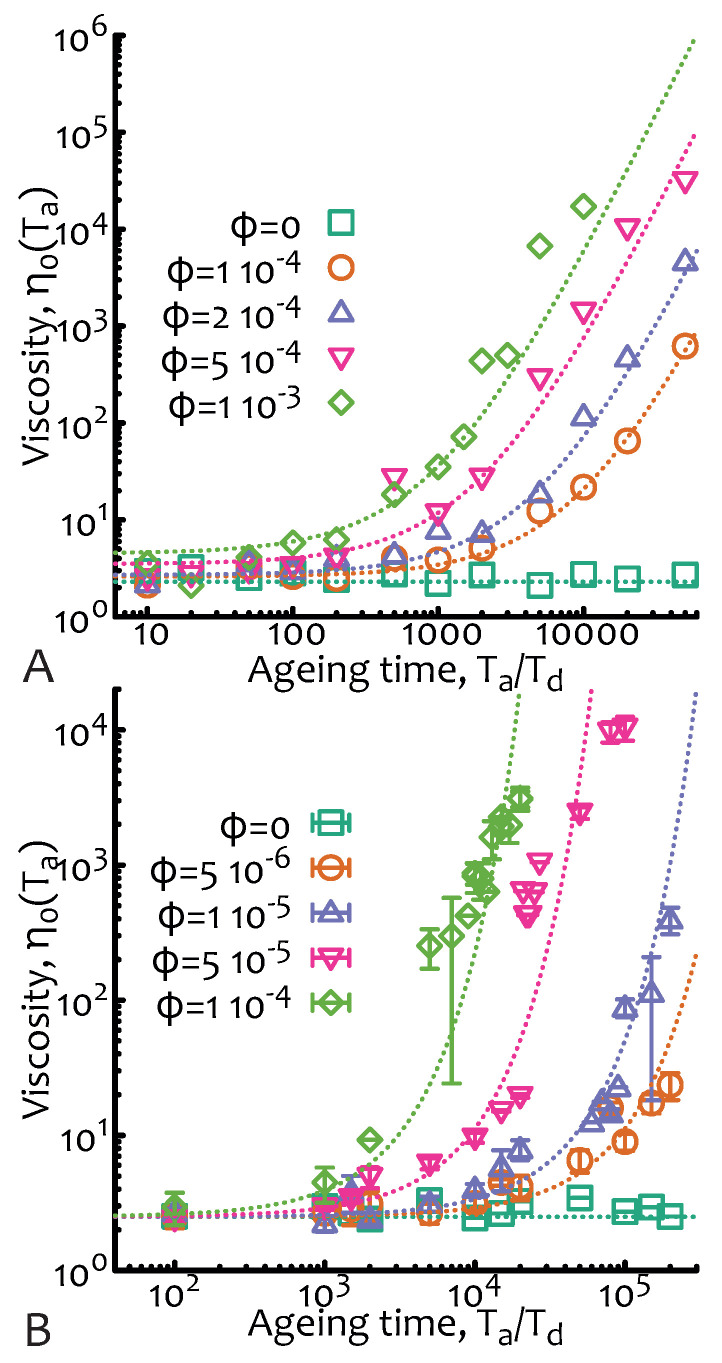
**Non-equilibrium living polymers—fusion only.** Zero-shear viscosity as a function of the ageing time $T_{a}/T_{d}$ and for different values of $\phi =T_{d}/\tau_{f}$. In (**A**) the case of a test chain within a system of non fusing chains is shown. Dashed lines are obtained from Equation (Equation 14) multiplied by a numerical pre-factor. In (**B**), the self-consistent case—in which all chains in the system grow at the same rate—is shown. Dashed lines are obtained from Equation (Equation 17) multiplied by a numerical pre-factor. In this case I am also showing error bars as the dispersion is larger than the other cases. The data points are from Brownian simulations as described in Figure 2A with no breakage and with parameters $D_{0}=0.1$ dL${}^{2}$/dt, dL=0.01, dt=0.01 and $L_{0}=1$.

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
