# Peer review of "Non-Equilibrium Living Polymers"

_entropy, 2020, doi:10.3390/e22101130_

Round 1

Reviewer 1 Report

Review in attachment.

Author Response

I thank the referee for carefully reading the manuscript and for her/his positive comments. I have now addressed all her/his suggestions for minor revisions as detailed below:

-Figures 2,3 and 4 provided charts including lines and points. I understanded that the lines present theoretical curves coming from presented equations. But it wasn’t clearly stated what is presented by the marked points on the all figures. Please consider expanding the legend on those figures and if there were simulations performed – to include more “behind the curtain” description of their parameters, etc.

I have added an explanation that the data points are from Brownian simulations of the numerical stochastic algorithm described in Fig2A and added numerical values for the parameters not specified in the figures. Please note that I also share the source codes for these simulations in a git repository as indicated in the acknowledgement section.

-Page 2, before Eq (3) a complex stress relaxation G*(ω) appears but is it not used later at all. The whole sentence starting from “In general, the stress...” is not clear. 

I reworded this part.

-Page 3: “Starting from the standard theory for equilibrium living polymers, there are several out-of-equilibrium generalisations that can be proposed and studied.” – I suggest to provide citations. I suppose that this Section presents the Author’s propositions of generalisations of the presented theories but it is hard to guess it reading the section.

I reworded this part and added a citation.

Also sentence: “The numerical implementation of this...” didn’t inform who is the author of the numerical implementation and how it was made.

The author is myself, and I provide the codes used in a git repository. I now specify this information whenever I refer to a "numerical implementation".

-Page 4: “It should be mentioned once more that since there is no counterpart to each architectural change, both systems are driven to an absorbing state in which all polymers are either broken down to individual segments or fused in a giant structure.” – why Author use “absorbing state” terminology here? I propose to expand on the idea to make it more clear.

In practical realisations, the sample is finite and contains a finite number of polymers. Hence, once they are all broken up or fused together, there is no way to get back from these states; hence, they are "absorbing". I now elaborate more in the relevant part of the text.   

Page 5: any citation telling about DNA and its restriction sites could be helpful. Also statement about “time for a typical digestion experiment” should have any citation.

I added a reference to a very good review on restriction enzymes.

Page 5: is NRS a number of restriction sites?

Yes. Added. thanks. 

Minor technical remarks:
- Page 3, below Eq (8) – “rather as...as...” – as->than? Please consider rewording this
phrase.
- And later: “ad random” -> “at”
- Page 3 – please consider using more fitting word than “counterpart” in “(without the
counterpart)” or expanding the phrase to indicate what counterpart you have in mind there.

All done. Thanks for flagging them. 

Reviewer 2 Report

This is a nicely written manuscript that should be published.
Using scaling arguments supported by numerical simulations, the author reviews the present state of
equilibrium living polymers and propose a non-equilibrium generalization.
Both theoretical and experimental perspectives are presented.

I am pleased to recommend publication in the present form.

Author Response

I thank the referee for her/his support.